# More Offenders, More Crime: Estimating the Size of the Offender Population in a Latin American Setting

**Spencer P. Chainey** * and **Dennis L. Lazarus**

Department of Security and Crime Science, University College London, 35 Tavistock Square, London WC1H 9EZ, UK; dllazarusg@gmail.com
* Correspondence: s.chainey@ucl.ac.uk

**Abstract:** Research that has examined the high levels of crime experienced in Latin American settings has suggested that macrostructural variables (such as social inequality), and factors associated with development and institutional capacity, offer explanations for these high crime levels. Although useful, these studies have yet to quantify how these explanations translate to the dynamics of offending activities. In the current study, we examine a key component related to offending dynamics: the size of the offender population. Using two capture-recapture techniques and a bootstrap simulation, estimates were generated of the sizes of the offender populations for three comparable cities in Brazil, Mexico, and England. Each of the techniques generated similar estimates for the offender population size for each city, but with these estimates varying substantially between the cities. This included the estimated offender population size for the Brazilian city being twenty-five times greater than that for the English city. Risk of arrest values were also generated, with these calculated to be substantially lower for the Brazilian and Mexican cities than for the English city. The results provide a quantification of criminal behavior that offers a potential new insight into the high levels of crime that are experienced in Latin American settings.

**Keywords:** offender population; capture-recapture; risk of arrest; impunity; Brazil; Mexico

## 1. Introduction

Countries in Latin America experience some of the highest crime levels in the world (van Dijk et al. 2021). Mortality due to violence is more than five times the rate in Latin America than it is in Western Europe (Soares and Naritomi 2010), robbery rates are higher in Latin America than in any other part of the world (Muggah and Tobón 2018), and Latin America is where forty-two of the fifty most violent cities in the world are located (Citizens Council for Public Security and Penal Justice 2019). To date, the majority of the research that has attempted to explain these higher levels of crime has focused on macrostructural explanations, such as social inequality and poverty (Baumer and Wolff 2014; Bergman 2009; Bourguignon et al. 2003; Chamlin and Cochran 2005; Fajnzylber et al. 2002; Hsieh and Pugh 1993; Koeppel et al. 2015; Machin et al. 2011; Messner 1989; Messner and Zimmerman 2012; Nivette 2011; Schultze-Kraft et al. 2018; Trent and Pridemore 2012; Vilalta et al. 2016). Alternative, but similarly macro explanations for higher levels of crime have included: rapid and disorganized urbanization (Neumayer 2003); population density (Nivette 2011); cultural masculinity (Neapolitan 1994); transition to democratic rule and political stability (Lafree and Tseloni 2006; Neumayer 2003; Rivera 2016); the illegal drugs trade (Snyder and Martínez 2017); the proliferation of arms trafficking and organized crime (Esparza et al. 2019; Garzón-Vergara 2016); the rule of law (De Boer and Bosetti 2015); governance and corruption (Chainey et al. 2021a); and the resources available to police agencies in Latin American countries compared to other regions of the world (Rivera 2016). Although research about these structural, developmental, and institutional factors have been useful for improving our understanding of the high crime rates in Latin American settings, these

studies have not adequately connected these macro conditions to the dynamics of criminal offending in these settings.

The level of crime an area experiences is related to three characteristics associated with the dynamics of criminal offending: the size of the offender population, the frequency with which each offender commits crime, and the duration of the offenders' criminal careers (Blumstein and Graddy 1982; Visher and Roth 1986). Limited research exists on these factors in Latin American settings and how they explain the high levels of crime in the region. When examining how offending dynamics influence variations in crime levels, the first step to take is to determine how many people participate in criminal behaviour (Blumstein and Graddy 1982). No known study has provided details on the size of the offender population in a Latin American setting. In the current study, we take this first step by estimating the sizes of offender populations in Latin American settings and consider if this may contribute to the high levels of crime in these settings. We return to discuss offending frequency and duration in a later section. We hypothesize that the size of the offender populations in Latin American settings are greater than that observed in a comparable setting.

Capture-recapture models have been used in many settings (but not Latin America) to estimate the size of the offender population. Capture-recapture models require data about offenders who have been arrested for committing a crime and how many times each of these offenders has been arrested. To date, estimates of the sizes of the offender populations in Latin American settings have not been calculated, and we believe this is associated with the paucity of arrest data that is recorded by police agencies in the region. To identify suitable study areas in Latin America, we consulted several police agencies in the region to determine whether data were recorded in the format that was required. We provide a commentary about the availability of the arrest data in the first part of the Data and Methods section.

After identifying suitable sources of recorded data, we estimated the offender population sizes in Latin American settings and compared the results to an English setting, comparing cities similar in size but with different crime rates. The study used two capture-recapture techniques and a bootstrap simulation approach to estimate the sizes of the offender populations, and to compare the estimates that each technique generated for each of our study areas.

Our study is novel. It is a proof of concept for estimating offender population sizes in Latin American settings rather than being a study that offers conclusions on the inferential relationship between crime levels and offender populations in Latin America compared to elsewhere. The approaches we use for estimating the sizes of the offender populations are illustrative of how this calculation is possible, as well as how it could be replicated for other Latin American settings and elsewhere. We anticipate that the estimates of the offender population sizes we generate can offer insights about offending dynamics in Latin American settings, and prompt new research that examines these offending dynamics in relation to macro explanations for high levels of crime.

In the next section, we elaborate on the objectives of the current study and describe the approaches for estimating the size of offender populations that led to the choice of techniques we use in the current study. We then describe the data and techniques we used, present the results, discuss the findings, and describe the limitations of the current study. Conclusions are provided in the last section.

## 2. Research Objectives and the Estimation of Offender Population Size

In this study, we examine if the size of the offender population is larger in a Latin American setting than it is in a Western setting. Although the macro conditions described in the previous section offer explanations for the variation in crime levels between different settings, the current study considers if offender population size is, in itself, a potential factor for why there are higher levels of crime in Latin American settings. The focus of the current study is the attempt to estimate offender population sizes in Latin America settings

and compare these findings to a non-Latin American setting. We do not analyze why there are differences in offender population size. However, we do speculate on the reasons for this after describing our results.

Most estimates of offender participation in crime rely on longitudinal cohort studies, or self-reporting criminal involvement surveys, that examine involvement in criminal activity for a representative sample of the population. This has led to the creation of approaches, such as the cumulative participation rate and the offending hazard rate, as measures to estimate the size or likelihood of the offending participation in a population (Visher and Roth 1986). Cohort studies of this type, or self-reporting surveys of criminal involvement in Latin American settings, are rare. For example, the International Self-Report Delinquency Study includes almost all European countries but includes only one of the thirty-four countries[1] in Latin America (Venezuela) (Marshall et al. 2015). Self-reporting surveys have, however, been questioned for their accuracy in determining offender participation in Latin American settings, where instrument design and use has been met by cultural challenges (Rodríguez et al. 2015). Victimization surveys have also been considered as a data source for estimating the size of offender populations. However, these have been discounted because they do not take into account the multiple offences of individual offenders and, thus, lead to an overestimation of the offender population (Bouchard and Lussier 2015).

Capture-recapture methods offer an alternative to cohort studies and self-reporting offending studies for estimating the size of a population. Capture-recapture methods originate from applications in biology and ecology to calculate the size of animal populations (Schwarz and Seber 1999). These methods have since been applied to calculate offender populations (Bouchard and Lussier 2015; Bouchard and Tremblay 2005; Rossmo and Routledge 1990; Tajuddin et al. 2021; van der Heijden et al. 2014). Capture-recapture methods use the recurrent pattern observed in a visible population to calculate the size of the population that is hidden from observation (Bouchard 2007). This involves examining patterns observed in data that are associated with offenders who were arrested to estimate the number of offenders who were not arrested. After calculating the size of the hidden population and adding this to the size of the population that is known (i.e., have been captured or, in the context of crime, offenders who have been arrested), the total size of the population can be estimated.

Longitudinal studies on criminal careers that have estimated the prevalence and incidence of arrest have shed some light on how being arrested at least once is not an uncommon occurrence (Tillman 1987). Arrest data provide a measure of the known offending population and can be used to establish an estimate of the likelihood of a person who has committed a crime being arrested (Bouchard and Tremblay 2005). This, in turn, can be used to estimate the total size of the offender population (Zelterman 1988). Bouchard and Tremblay (2005) describe how capture-recapture techniques can calculate the likelihood of arrest by defining a population of *susceptibles* who have not been arrested, but who nonetheless have the same characteristics as those who have been arrested. This population of susceptibles refers to individuals who have been active in crime commission, but who have been hidden because they have not been captured for an observation period for which data on other individuals who have been arrested is available. Patterns of re-offending for those individuals who have been arrested are assumed to be shared with only a small proportion of the full offending population (of both those known and those hidden) for any given observation period (Bouchard and Tremblay 2005). The likelihood of arrest is not defined as the likelihood of being arrested for each act committed but, rather, as a measure of the overall outcomes for individuals who display similar offending patterns as those who have already come into contact with the police. This means the likelihood of arrest is not associated with a particular individual and is unlikely to be affected by improved tactics used by police agencies to try to arrest more offenders. Therefore, arrested offenders are not considered to possess any particular characteristics that led to their arrest but, rather, are considered to operate in the same manner as those who have not been arrested

and, therefore, are representative of the full population of active offenders (Bouchard and Tremblay 2005).

Capture-recapture techniques for estimating offender populations use recorded police data on arrests to calculate the estimated offender population. Although some researchers have expressed concerns about the validity of arrest data as a sampling method for determining the size of the criminal population (Blumstein and Cohen 1979), the reasoning behind this concern is based on conjectures about the behavior of individual criminals. Although there can be a degree of variation in the frequency of encounters that individual offenders have with the police, there is no evidence to suggest the existence of some form of super-offender that evades detection for extremely long periods of time, while at the same time engaging in criminal activity at a high level (Bouchard and Tremblay 2005). This same assumption is made in the field of biology, where super-animals within a population of the same animal, constantly successful in evading observation, are not considered to exist, hence leading to an underestimation of the size of the animal population. Additionally, other studies have shown that the arrested population for certain crimes, such as those that involve face-to-face contact between the offender and the victim, is almost identical in its demographic characteristics to the population of offenders reported by victims in victimization surveys (Hindelang 1976; Wittebrood and Junger 2002), and that the arrest frequency across ethnic groups is proportionate to an ethnic group's involvement in criminal activity (D'Alessio and Stolzenberg 2003). Hence, arrest data are considered to be suitably reliable sources of information that are representative of the whole offending population and, in particular, for crimes that involve personal contact between the offender and victim (Bouchard and Tremblay 2005). Additionally, a benefit of using arrest data is that these data are determined by their involvement with two institutions: the police agency that arrests the offender, and the criminal justice system that defines the criminal code for the activity that warrants an arrest. This can reduce bias towards a single viewpoint associated with why a person was arrested. Capture-recapture techniques also avoid the limitations of the cultural or cognitive differences that are inherent in standardized survey instruments, and so provide a more accurate method for comparing cross-national estimations of offender population sizes.

In the current study, we use arrest data recorded by police agencies, and apply these data in capture-recapture models to estimate the size of offender populations. The use of capture-recapture techniques for estimating offender population size using arrest data involves creating an offender-arrest count frequency table that is then input into the capture-recapture model. The basic premise behind these techniques is to use information about offenders, who have been arrested once, twice, three, to n times (i.e., the known population of offenders), to calculate the number of offenders caught zero times (i.e., the hidden population of offenders). The distribution of these count frequency tables follows a Poisson distribution, from which it is determined that offenders commit crime at a Poisson rate (Nagin and Land 1993). That is, offenders do not necessarily need to have a constant probability of arrest but, rather, have a susceptibility to being arrested given that they engage in criminal activity.

Capture-recapture models generate more accurate estimates when applied to estimating the offender population size for certain types of crime, rather than all crime more generally (Larson et al. 1994). Hence, in the current study, rather than estimating the offender population size for offenders who have committed any type of crime, we estimate the offender population size for offenders who have committed a specific type of crime: robbery from the person. We selected this type of crime because robbery from the person takes place in circumstances that are similar in most settings and is similar in its commission in these settings (i.e., it involves an offender using the threat or act of violence to steal an item of property from a person that the person is likely to possess, e.g., a cell phone or a wallet/purse containing money or bank cards). Additionally, the police recording definitions for robbery from the person are the same in the countries included in the sample (UNODC 2017) and are types of crime that are reported to the police more than

any other crime types that involve face-to-face contact, such as assaults (INEGI 2019; Home Office 2018). In addition, levels of robberies from the person are higher in Latin American settings than they are in North American and European settings (UNODC 2017), allowing for the study to speculate on whether these differences are associated with differences in offender population sizes. From this, and through the use of capture-recapture techniques, we hypothesize that the estimated size of the offender population for robbery in a Latin American setting is greater than that observed in a comparable setting where robbery levels are lower.

## 3. Data and Methods

In this section, we begin by providing a commentary of the consultation we conducted that resulted in us selecting the research areas for the current study. Capture-recapture models require arrest data that uniquely identifies each individual who has been arrested for a crime, with this identifier being used to identify the same person if they are arrested multiple times. This identifying information can include the name of the offender, but it is more accurate when each offender is assigned a unique identifying number that is used for each subsequent occasion that the same offender is arrested. Arrest data that are recorded by police agencies in several countries uses unique identifying numbers for each offender that is arrested. We had already sourced arrest data for the city of Newcastle, England because of our involvement in a research project with the police agency in this city. Hence, our inclusion of Newcastle as a study area was because of convenience, and our familiarity and access to these data. The offender arrest data for Newcastle contained a unique identifying number for each offender. In police agencies in Latin America, recording a unique identifying number for each offender is less common. The first part of the current study involved consulting police agencies in Argentina, Brazil, Chile, Colombia, El Salvador, Mexico, and Uruguay for arrest data recorded in the required format. These countries were chosen because they had more advanced police data recording standards than other countries in Latin America (Bergman 2018), and partly because of convenience as the authors were able to utilize contacts with police agencies in these countries.

Offender arrest data that contained unique identifying numbers for each person who was arrested was not recorded by the police agencies we consulted in Argentina, Chile, and El Salvador, limiting the potential use of offender arrest data to Brazil, Colombia, Mexico, and Uruguay. An additional challenge in sourcing data for the research was obtaining the permission that was required by each police agency to allow for the use of sensitive offender arrest data because of the data's personalized content, and the transfer and storage of the data for research use. The ability to share data for the purposes of the research further limited us to using samples from only Brazil and Mexico. We were able to set up data sharing agreements and the secure electronic transfer of offender arrest data with Brazil and Mexico, but were unable to make similar arrangements with the police agencies in the other countries within the time period that the research was conducted. We then decided to extend our data search across Central America to Belize, where data in the necessary format was made available, but the size of the sample was too small for generating accurate estimations using capture-recapture techniques. We return to the issue of sample size in a later section. At this point, we decided to focus our efforts on using data from Brazil and Mexico, and on offenders arrested for robbery from the person for the reasons explained in the previous section.

The data selected for analysis was further narrowed down by being selected from cities that were most similar in size and function to Newcastle, and from the areas within these countries where we had the best police contacts. This resulted in choosing data for two cities: Cuiabá in Brazil, and Escobedo in Mexico. We re-emphasize that the aim of the current study was not to provide inferential evidence on the relationship between offender population sizes and crime levels in Latin America as a whole but, rather, was an attempt to estimate the offender population size in a Latin American setting, demonstrate how this can be done, and use the findings to offer new insights that could quantify the potential

influence of the offending dynamics on the high levels of crime in Latin American settings. The sample of three cities was sufficient for our intended research aims. Each of the cities were medium-sized cities, with populations between 300,000 and 600,000, and each city was similar in its urban function (such as containing a mix of residential and commercial areas and offering services and entertainment). Arrest data on robbery from the person for each city were extracted from police recording systems for a one-year period in 2018. A one-year period of data was considered to be sufficient because it replicates the sample size used in similar studies for estimating offender population size.

Capture-recapture techniques require arrest data on individuals to be aggregated to determine the number of individuals caught once, twice, and n times for the observation period (Wilson and Collins 1992). A number of capture-recapture techniques have been developed for estimating the size of a population. The current study used three techniques for estimating offender population size: Zelterman's zero-truncated Poisson estimator; a simple fitted Poisson model; and a bootstrap resampling simulation technique. Three techniques were chosen so that we could compare the results that each produced for each study area. The three techniques we chose were selected because of their proven accuracy in estimating hidden populations (Brittain and Böhning 2009).

Zelterman's truncated Poisson estimator (Zelterman 1988) is given by Equation (1):

$$\hat{N} = \frac{N}{1 - e^{\left(-2 \times \frac{n_2}{n_1}\right)}} \tag{1}$$

where $\hat{N}$ is the estimate for the total offender population, $N$ is the total number of individuals arrested, $n_1$ is the number of individuals arrested once, $n_2$ is the number of individuals arrested twice, and $e$ is a constant. One of the advantages of this technique is its relative simplicity, with estimates being straightforward enough to calculate once the data have been arranged into counts for offenders arrested once, or at least twice. The technique assumes that offenders caught once or twice, rather than those who have been arrested three of more times, are representative of patterns of offending that are closer to those arrested zero times. As it is not uncommon for individuals who offend to be arrested at least once (Tillman 1987), it is then justified to assign a greater weight to offenders who offend at least twice, as these individuals are disproportionately responsible for the observed level of crime. Zelterman's truncated Poisson estimator also diminishes the influence of outliers that have been arrested an inordinate number of times, and that skew the homogeneity of the offending population (Bouchard 2007).

The second technique used was the fitted Poisson model. The fitted Poisson model applies an expected Poisson distribution to the arrest data. As stated in the previous section, the counts of the number of times an individual is arrested tends to follow a Poisson distribution. However, the observed data for arrests naturally do not include the count for offenders caught zero times. Thus, the observed data follows a zero-truncated Poisson distribution. If the lambda ($\lambda$) (parameter, or rate) is estimated with $\hat{\lambda}$, from this the total offender population can be calculated by normalising the observed arrests with their corresponding estimated probability mass. To do so requires $\lambda$ to be found, which is the estimate of the missing observations. The problem then is to estimate the $\lambda$ that specifies the distribution. To this end, a generalized linear model (GLM) with a logarithm of $\lambda$ as a link function can be used. Notice that, in the absence of covariates, this problem is equivalent to fitting a zero-truncated Poisson distribution to the observed data, and solving the problem numerically is equivalent to fitting a GLM composed only of an intercept parameter. This means that only the intercept value needs to be calculated, which is possible from fitting a GLM. Again, because there are no covariates, $\hat{\lambda}$ specifies the law for all the observed data and suffices to compute the total population. Additionally, the variance of the estimator enables the calculation of a confidence interval for $\hat{\lambda}$ and, correspondingly, for the estimated population. We describe below how we computed this confidence interval measure.

The bootstrap resampling simulation technique involves using sampling replacement to generate offender population estimates. The purpose of using this technique was to

attempt to simulate the process that generated the original datasets. The technique involves creating multiple randomized replacement datasets that are the same size as the original arrest dataset for each city. A total of 5000 simulations were run using a sampling replacement for each arrest dataset. For each simulation result, a Poisson distribution was fitted using a pointwise estimator to determine an estimate of the offender population size. The final estimate of the offender population size using this bootstrap resampling simulation technique was taken as the mean of the estimates generated from the 5000 simulations. The reason for running 5000 simulations was arbitrary but was considered to be sufficient for generating reliable estimates of the offender population size using this method. To examine the consistency of the estimates of the offender population size using 5000 simulations, we repeated the use of the technique using 1000 simulations and 10,000 simulations, and no differences were observed in the estimates that were generated.

For each technique, confidence intervals for the estimation of offender population sizes were generated. For the calculation of the confidence intervals for the Zelterman method, we used the variance of the estimations (following the calculation of confidence intervals used by Bouchard 2007). For the simple fitted Poisson model, we used the confidence intervals of the λ parameter (following van der Heijden et al. 2003). For the bootstrap resampling method, we obtained an empirical distribution of the parameter λ from the 5000 simulations and used the percentiles of 0.025 and 0.975 to generate confidence intervals. We were not able to calculate the confidence intervals for data from Escobedo using the bootstrap method because the sample size was too small and the frequency of ones was too high (i.e., most resampling datasets that were generated also mainly contained ones). We return to this limitation in a later section.

The Zelterman's truncated Poisson estimates and confidence intervals were calculated in Microsoft Excel by applying Equation (1) to the data. Estimates generated from the simple fitted Poisson model technique (and confidence intervals) were calculated using the *GLM* package in R (R Core Team 2020) and the bootstrap resampling simulation technique was applied using the R package *boot*. In addition to estimating offender population size, the outputs from the techniques were used to calculate the likelihood of arrest for offenders (of robbery from the person) in each of the three cities. To calculate the likelihood of arrest we applied Equation (2):

$$likelihood\ of\ arrest = \frac{total\ number\ of\ individual\ offenders\ arrested}{total\ number\ of\ estimated\ individuals} \qquad (2)$$

## 4. Results

Table 1 shows the differences in the number of robberies from the person for each city. Cuiabá and Escobedo recorded more offences than Newcastle but were larger cities. The robbery rate for Cuiabá was four times greater than that for Newcastle. Escobedo had a crime rate that was forty-eight percent greater than that for Newcastle.

**Table 1.** Population and robberies (n and rate) in 2018.

| City | Population | Robbery from the Person (Rate per 100,000 Population) |
|---|---|---|
| Cuiabá, Brazil, | 425,148 * | 2057 (484) |
| Escobedo, Mexico | 607,153 ** | 1051 (173) |
| Newcastle, England | 300,196 *** | 350 (117) |

Source: * Instituto Brasileiro de Geografia e Estatística (2019); ** Instituto Nacional de Estadística y Geografía (2019); *** Office for National Statistics (2019).

Table 2 shows the number of offenders arrested by the number of times they were arrested for robbery from the person in each city during the observation period. Even though Escobedo was the largest of the three cities, it recorded the lowest level of offender arrests, with no offenders arrested more than twice. Cuiabá was more similar in population size to Newcastle, but the recorded number of offenders arrested was four times greater.

However, in Newcastle more offenders were arrested multiple times for robbery from the person. These arrest counts are also shown in Figure 1, with each city showing a Poisson distribution for arrest counts.

**Table 2.** Number of offenders arrested, by the number of times they were arrested for robbery in Cuiabá, Escobedo, and Newcastle.

| Arrested | Cuiabá | Escobedo | Newcastle |
|---|---|---|---|
| Once | 661 | 57 | 104 |
| Twice | 29 | 2 | 46 |
| Three times | 3 | 0 | 16 |
| Four times | 1 | 0 | 4 |
| Five times or more | 0 | 0 | 0 |
| Total number of offenders arrested | 694 | 59 | 170 |

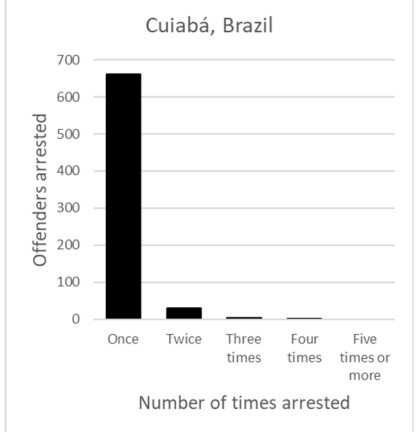 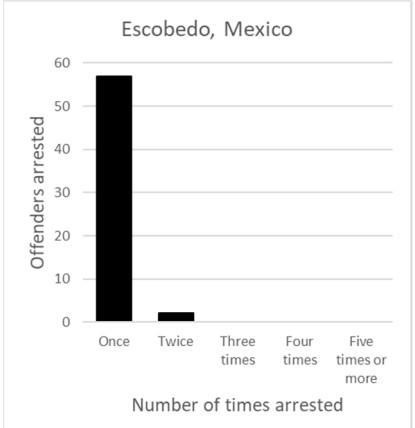 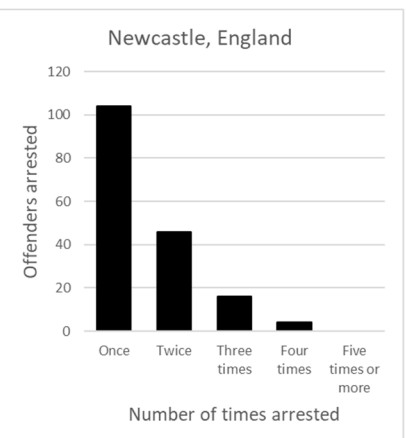

**Figure 1.** The distribution of offenders arrested by the number of times they were arrested for robbery in Cuiabá, Escobedo, and Newcastle.

Table 3 lists the results from the three techniques that were used for estimating the size of the offender population. These results include the estimate for the number of offenders in the hidden population (who were active at least once during the observation period but evaded capture) and the estimate for the full offender population size. The Zelterman truncated Poisson technique estimated that 7567 offenders evaded capture (for robbery from the person) in Cuiabá, compared to 812 in Escobedo, and 120 in Newcastle. For Cuiabá, the estimated total offender population size was calculated as 8261—twelve times greater than the arrested population of 694 for the observation period. In Escobedo, the estimated offender population size was 871—fourteen times greater than the number of offenders who had been arrested in this city. In Newcastle, the difference between the estimated offender population size and the number of offenders arrested was much lower—it was estimated there were 290 active offenders during the observation period, just under double that of the number of offenders who had been arrested. Of particular note was that even though Newcastle had originally recorded three times more arrests than Escobedo for the observation period, the estimate for the hidden offender population in Escobedo was almost seven times greater than the hidden offender population in Newcastle. Similarly, although there were four times more offenders arrested in Cuiabá than in Newcastle for the observation period, the estimated offender population size in Cuiabá was 28 times greater than that for Newcastle. We also note that the estimated size of the offender population in Cuiabá and Escobedo was greater than the number of robbery offenses that were recorded by the police for the time period of analysis (see Table 1). We return to these results in the Discussion section when we consider other research that has examined differences

in the *dark figure* of crime (i.e., the difference between police recorded crime and actual occurrences of crime) between Latin American countries and the United Kingdom.

**Table 3.** Estimates for hidden population, offender population size and risk of arrest using the Zelterman, Fitted Poisson and Bootstrap techniques.

|  | Cuiabá | | | Escobedo | | | Newcastle | | |
|---|---|---|---|---|---|---|---|---|---|
|  | Zelterman | Fitted Poisson | Bootstrap | Zelterman | Fitted Poisson | Bootstrap | Zelterman | Fitted Poisson | Bootstrap |
| Estimated hidden population | 7567 | 6110 | 6209 | 812 | 851 | 28,365 | 120 | 113 | 114 |
| Estimated offender population size | 8261 | 6804 | 6903 | 871 | 910 | 28,424 | 290 | 283 | 284 |
| Confidence interval | 6893–10,308 | 5062–9193 | 5099–10,098 | 518–2722 | 253–3521 | NA | 261–325 | 253–320 | 255–324 |
| Estimated risk of arrest | 8.4% | 10.2% | 10.1% | 6.8% | 6.5% | 0.2% | 58.7% | 60.1% | 59.9% |

The offender population size estimates in Table 3 are also listed with confidence intervals. These show that, for the Zelterman estimator, the confidence intervals were reasonably narrow for Cuiabá and Newcastle, but wider for Escobedo. Capture-recapture models require data on offenders who are recaptured for them to be accurate. Arrest data from Escobedo mainly referred to offenders who were caught only once, which in turn results in the generation of wider confidence intervals. We consider the impact of limited recapture information about offenders for estimating offender population size, and on the calculation of confidence bands, in the Discussion section.

Table 3 also lists the estimated risk of arrest for robbery from the person, calculated from the arrested population and the estimated offender population size. For Cuiabá and Escobedo, the risk of arrest was similar (8.4% and 6.8%, respectively). In Newcastle, the risk of arrest was much higher (58.7%). These results indicate that even though Escobedo recorded fewer arrests than in Newcastle, this may not have been because of fewer offenders operating, but may have been because of fewer detections for robbery from a larger offender population in Escobedo. We return to these results, with respect to differences in impunity, in the Discussion section.

The hidden offender population size estimates that were generated using the fitted Poisson model were of similar magnitude to those calculated using the Zelterman technique; the fitted Poisson model estimates were: 6110 in Cuiabá; 851 in Escobedo; and 113 in Newcastle (see Table 3). Confidence intervals, however, were larger for the fitted Poisson model than they were for the Zelterman estimates. The results from the fitted Poisson model generated similar estimates for the risk of arrest for robbery to those calculated from the Zelterman estimate for each city. The fitted Poisson model risk of arrest results were 10.2%in Cuiabá, 6.5% in Escobedo, and 60.1% in Newcastle.

The bootstrap resampling simulation technique generated similar hidden offender population estimates to those generated using the two other techniques, with the exception of Escobedo: 6209 in Cuiabá; 28,365 in Escobedo; and 114 in Newcastle (see Table 3). The result for Escobedo highlights the volatility in estimating hidden offender population sizes when arrest data contains limited information about the offenders who were recaptured. The confidence interval for the estimates using bootstrap simulations were larger for Cuiabá than for those generated using the Zelterman estimator, but were slightly smaller for those generated for Newcastle. Confidence intervals could not be generated for Escobedo using bootstrap simulation because of the high frequency of ones in the offender-arrest count frequency data that prevented a resampling of datasets that were different to that which had been observed (i.e., all simulated datasets mainly contained arrest data that related to offenders only being arrested once).

Figure 2 shows the estimated offender population size generated from each technique for each city (with the exception of Escobedo where the results from the bootstrap simulation technique are not included because of the volatility in this result). These figures illustrate the similarity in the offender population sizes calculated for each city from each technique and highlight the differences in estimated offender population sizes between the three cities. Taking the mean from the techniques, the estimated offender population size for robberies in Cuiabá was 7323, in Escobedo it was 890, and in Newcastle it was 285. Figure 2a–c also highlight the differences in confidence intervals between each city, with these being greatest for Escobedo because of the limited information on offenders who were recaptured.

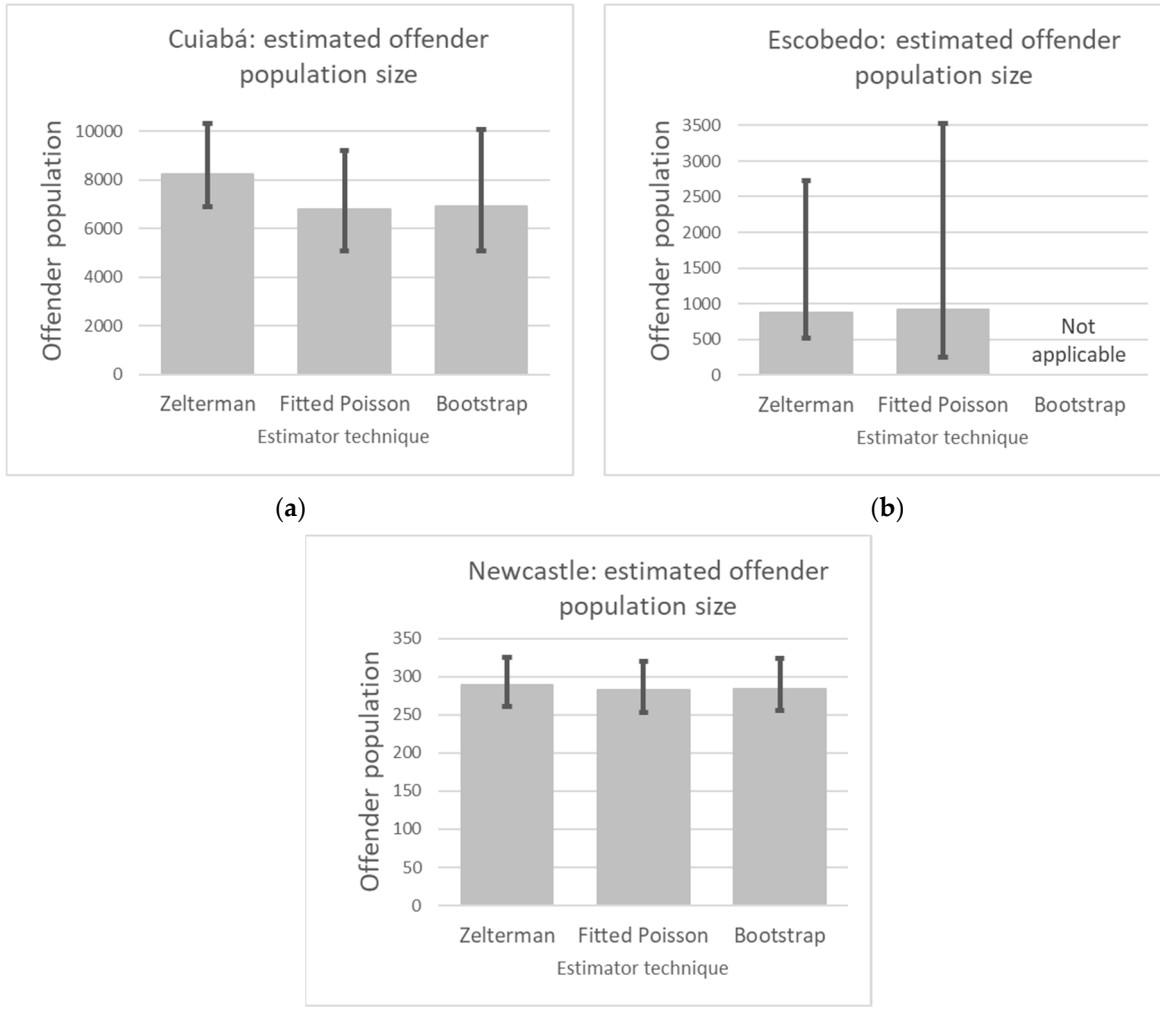

**Figure 2.** Estimated offender population sizes for robbery from the person in (**a**) Cuiabá, (**b**) Escobedo, and (**c**) Newcastle.

## 5. Discussion

To date, most studies that have examined reasons for the high levels of crime in Latin America have focused on stating how structural variables, such as inequality and poverty, and factors associated with development and institutional capacity, are key explanatory variables. Although these studies have been useful, these variables offer macro explanations for the high levels of crime and have not quantified how these variables result in the commission of more crime. In the current study, our aim was to take this reasoning forward by estimating the size of offender populations in Latin American settings.

After consulting with several police agencies in the Latin American region, two study areas (Cuiabá, Brazil, and Escobedo, Mexico) were selected where arrest data were available in the required format and could be used to estimate offender population sizes. Cuiabá recorded the highest levels of robbery from the person between the three cities, followed by Escobedo, with Newcastle recording the lowest. For Cuiabá, the estimated size of the offender population for robberies from the person was 7323 (based on a mean of the three techniques used), and for Escobedo, the mean estimated size of the offender population was 890. For Newcastle, the average estimated size of the offender population was smaller than for the other two cities: 285. Although Cuiabá was the largest of the three cities (by a magnitude of approximately twice the size of the other two cities), the estimated offender population in Cuiabá was over 25 times greater than that for Newcastle. The estimated offender population size in Escobedo was over three times greater than that for Newcastle. In returning to our main stated hypothesis—that the estimated size of the offender population for robbery in a Latin American setting is greater than that observed in a comparable setting where robbery levels are lower—the results provide an indication that the size of the offender population in Latin American settings is greater than those in other settings.

Collectively, the results from the current study suggest the likely presence of an association between the size of the offender population and the level of crime that an area experiences. The rank order for the size of the offender populations in each city followed the order of the crime levels experienced in these cities, and the differences in magnitude between offender size and crime levels were on par between each city. We do, however, note caution with reading too far into these estimates because of the small sample size of the cities (which we discuss further in the Limitations section), but we believe the findings meet the objectives of the proof-of-concept exercise we aimed to complete. We also take confidence in the estimates that were calculated because of the similarity in the estimates between the three techniques (with the exception of the bootstrap simulation results for Escobedo). We avoid calculating a ratio between the estimated offender population and the number of crimes because of the likelihood of differences in the reporting of crime by victims of robbery between the cities. In Latin American settings, the dark figure of crime (the difference between crime that is committed and crime that is reported) is much greater (Jaitman and Anauati 2020) than it is in settings such as Newcastle, so a ratio of the offender population and recorded crime could be misleading.

We noted in the Results section that the number of robberies recorded by the police in Cuiabá and Escobedo was less than the estimated size of the offender population for this type of crime. Expanding further on research findings about the dark figure of crime, it has been estimated that the dark figure for robberies in countries in Latin America was between 94% and 96% in 2014 (Jaitman and Anauati 2020) in comparison to a dark figure of 67% for robberies in the United Kingdom in the same study. This suggests that if only 4% to 6% of robberies were reported to the police in Cuiabá and Escobedo, the number of robberies that were actually committed by offenders in each city was more likely to be about 40,000 and 20,000 robberies, respectively. Hence, the estimated number of offenders calculated for each city was not greater than the likely number of robbery offenses that were committed in each city.

We encourage replication studies to confirm and build on our results. The results from the current study indicate that, in Latin American settings, the size of the offender population is larger than in most other settings. We also encourage replication of our study in other settings that experience high levels of crime, and where estimates of offender population sizes have not been calculated (e.g., countries in Africa) so that these studies can also improve our understanding about variations in offending dynamics and crime levels. When combined with estimates of the actual number of robberies committed, the results from the current study also provide an indication of the level of impunity that offenders in Latin American settings may be afforded; this is a point we further discuss below. Additionally, and in the wider context of social science research, we encourage

social scientists who examine the influence that structural conditions (e.g., inequality), development, and institutional explanations (e.g., government effectiveness) have on society to consider how these factors relate more directly to the behaviour of individuals. This is often lacking in social science research and, in particular, in research that straddles social science and criminal justice. For example, when researchers reveal significant relationships between variables representing structural conditions and variations in crime levels in statistical models, the validation of these associations would be more robust if each of the significant variables they identify is supported with an explanation about how they influence offending behavioural dynamics, such as the size of the offender population.

In the current study, the estimated offender population size calculated for Escobedo, using the bootstrap resampling simulation technique, was substantially different to that calculated using the two other techniques. This was because the offender-arrest frequency count data for Escobedo mainly consisted of ones (i.e., almost all offenders who were arrested in Escobedo were not rearrested), with this then limiting the random match sampling of data for simulation to similarly consist of mainly ones. In situations where the number of offenders arrested is small, this also creates difficulties in estimating the size of the offender population, particularly when this sample also consists of offenders who were only arrested once. This was the case with the data sample for Belize (the number of offenders arrested for robbery was 46), and this is why we did not include Belize in the study. The presence of a high number of ones in offender-arrest frequency count data also results in wide confidence intervals being generated, as illustrated in the other estimator results for Escobedo (see Table 3). Tajuddin et al. (2021) introduce a solution for dealing with offender-arrest count frequency data containing a high number of ones using a Horvitz–Thompson (HT) estimator under a one-inflated positive Poisson–Lindley model. Although the estimate of an offender population size using this model generates results that are similar to the capture-recapture models used in the current study, the benefit of the HT estimator is that it can generate narrower confidence bands (Tajuddin et al. 2021)[2]. We recommend the use of this estimator, alongside other capture-recapture models, in future studies that estimate the size of the offender population.

Capture-recapture models have certain assumptions. We discuss these in more detail in the Limitations section but, in general, any assumptions that a model possesses can undermine the accuracy of the model if there are questions about the validity of the assumptions. One particular assumption associated with capture-recapture models is that those arrested are considered to operate in the same manner as those who have not been arrested, so they are representative of the full population of active offenders. While some studies have examined this assumption, suggesting that the assumption is valid, these studies have been limited in the settings where they have been applied and where other theoretical factors associated with criminal behaviour require consideration. For example, none of these studies are based on samples of offenders in Latin American countries. Therefore, it is possible that, in different settings, certain assumptions may be less valid which, in turn, influences the validity of the results that capture-recapture models generate. While we encourage research that compares how offenders behave in different international settings, to validate the estimates of offender population size that can be generated from capture-recapture models, we encourage researchers to consider the use of alternative methods, such as the cohort studies and self-reporting offending studies that we described in Section 2, as well as any new capture-recapture models that are developed. By doing so, this will allow for results that are generated from different techniques to be compared and will help to examine if the differences in results relate to the assumptions associated with each model.

Using the results from each offender population estimation technique, an estimated risk of arrest value was calculated. In Cuiabá and Escobedo, the average risk of arrest (9.6% and 6.65%, respectively) was substantially lower than that for Newcastle (59.6%). Estimating the risk of arrest is beneficial because it can provide an indication of the effectiveness of police agencies in capturing offenders. Additionally, this value can offer an indication of

the perception offenders have of being caught for the commission of crime. Several studies have previously shown that impunity levels are higher in Latin America than they are in any other regions of the world (Bergman 2018; Nadanovsky and Cunha-Cruz 2009). For example, in Brazil, less than 10% of all homicides are solved (UNODC 2019), compared to over 87% in England and Wales (ONS 2018). If offenders, or would-be offenders, believe the risks of being caught and prosecuted for committing a crime are low, they are more likely to engage in criminal activity (Piquero and Rengert 1999; Wright et al. 2004).

Perceptions of being caught and prosecuted for committing a crime are associated with the theoretical principles of the rational choice perspective (Clarke and Felson 1993; Cornish and Clarke 1986), prospect theory (Tversky and Kahneman 1992), and deterrence. Rational choice and prospect theory offer ways for considering how an offender's interaction with the immediate environment influences the activity of crime commission (Chiu et al. 2011; Cornish and Clarke 2011). At their core is the concept that criminal behaviour is purposive, albeit bounded by the circumstances and situations within which offenders operate (Cornish and Clarke 2011) as well as the perceived gains from the commission of crime (Tversky and Kahneman 1992). Offenders, as the theories state, must make decisions by estimating the possible costs and benefits associated with their actions, with these costs and benefits being constrained by the risks or losses from being involved in criminal activity. These principles are also closely linked to deterrence theory, which emphasises that individuals are profit maximisers who consider the benefits of criminal activity against the perceived costs (such as risks, losses, threats, and efforts) involved. Deterrence theory asserts that people are discouraged from committing crimes if they believe there is certainty in the likelihood of being caught, and that punishment will be swift and severe (Kennedy 2009; Nagin 1998; Paternoster et al. 1983; Sherman 1993). In settings where the risks of being caught and prosecuted are low, and where individuals perceive the rewards from criminal activity to be beneficial, more individuals are likely to engage in criminality (Piquero and Rengert 1999; Wright et al. 2004). Hence, with these theoretical principles in mind (rational choice, prospect theory, and deterrence theory), although structural variables (and other variables discussed in this article) may influence why a person decides to become involved in criminal activity, in the situational settings where crime opportunities are available, if the risks of being caught are low, offending behaviour is more likely. If many individuals observe that the risk of offending is low and opportunities to commit crime are plentiful, this will likely lead to more individuals committing acts of crime. What is likely to result from this is an offender population size that is larger than in settings where the risk of arrest is greater.

Although estimating the size of the offender population can be useful for examining differences in crime levels between areas, this measure can also be used for a number of other applications. Estimating the size of the offender population that has not been arrested and has, thus, successfully avoided police detection practices, is likely to be information that is of value for identifying deficiencies in these practices and how they can be improved. Moreover, calculating and regularly updating the estimate of the size of the offender population for different types of crime can provide a useful measure for monitoring how this population size changes over time. For example, on occasions when changes in crime levels are observed, an understanding of whether the size of the offender population is associated with these trends could inform operational resource allocation decisions. Knowing the estimated size of the offender population, particularly when the estimate indicates that the offender population size is large, can also be used for informing interventions that specifically seek to reduce the offender population size. Several types of interventions aim to decrease crime by deterring offending behaviour. These include focussed deterrence programs (Braga et al. 2019), and hot spot policing (Braga and Weisburd 2010; Chainey et al. 2021b). Estimating the offender population size before and after the implementation of an intervention can be useful as an impact evaluation measure to observe how the offender population size has changed over time. Measuring the change in the offender population size can also be used for helping to

determine the specific mechanisms through which an intervention had an effect, such as whether the intervention did have a deterrent effect by reducing criminal participation. Moreover, when changes in policy are made, such as changes in the way the criminal justice system processes offenders (e.g., the change from an inquisitorial to adversarial system that has taken place in several Latin American countries in recent years), measuring the impact that a change in policy has on the size of the offender population can be useful.

An indication of the size of the offender population can also inform agent-based models (ABMs) that seek to simulate criminal behaviour. To date, ABMs include agents that replicate the behaviour of offenders (see Groff et al. 2019 for review of ABM applied to crime) but are often limited by not knowing how many offenders should be activated within the model and how these should vary between the different settings that a model simulates. Determining an estimate of the number of offenders to activate in an ABM for each setting would likely improve the accuracy of these models. Offender population size estimates can also provide meaningful information about police workload (van der Heijden et al. 2003) and can be used for determining the allocation of police resources with a higher degree of precision. The results from the current study also illustrate the value of police agencies recording data in the format required for estimating offender population size. We encourage police agencies that do not currently record data in the format required for capture-recapture models to implement practices that will allow them to calculate estimates of offender population size, and to then benefit from the ways these estimates can be used.

## 6. Limitations

Several assumptions are made about the population that is to be estimated using capture-recapture models. These relate to closedness, homogeneity, independence, and constant behavior. Closedness refers to the assumption that there is no growth in the size of the population during the period that the sample data relate. Limiting the period of time for which the data sample is taken (e.g., one year) helped to minimize changes in population size due to subjects (e.g., criminals) moving in or out of the population. The population under study is also assumed to be homogenous, which in offending terms relates to its demographic makeup. It is also assumed that the observations are independent. This means that the probability of a person being arrested is not related to the probability of another person being arrested. In reality, offenders often co-offend, with the arrest of one offender likely to influence the behavior of any co-offenders and them attempting to avoid capture. Constant behavior is also assumed. When capture-recapture studies are used in ecology, the change in behavior of the animal population in response to trapping is often considered and additionally built into models. In relation to offenders, offenders are likely to always consider the risk of capture in some way, and hence modelling deviations from a constant of behavior is required less than applications in ecology. However, the topics relating to these assumptions have received limited study in criminology research and are a recommended area for further research into improving how capture-recapture models can be applied in studies for estimating offender population size. Assumptions associated with statistical modelling techniques do mean that these can impose limitations and influence the accuracy of estimates that these techniques generate. In the current research, we attempted to address this by using two capture-recapture techniques to generate estimates, and a bootstrap resampling simulation approach, from which estimates that were generated could be compared and, if similar, the results generated from one technique would offer some validation of the results generated from the others.

The current study involved three cities—two from Latin America and one from England. Several other police agencies in the Latin American region were consulted and did record data in the required format for estimating offender population size. These agencies were not able to share these data because of the sensitive personal content and issues with securing their electronic transfer. Checks were performed on the data that were used to ensure they provided an accurate account of the known offender population, based on the offenders arrested. This included checking for data completeness and consistency

in the regularity of arrests (e.g., checking whether there were periods of time in the arrest data for each city when no offenders were arrested). No issues were identified and the data were considered to be complete. However, we recognize that, as we had no control over how the original data were recorded, issues with the data that we did not identify (such as the failure to record all arrested offenders, or the incorrect use of offender identification numbers) may be present and could have influenced the results. As previously stated, our research was, in part, a proof-of-concept exercise to illustrate that the estimation of offender population sizes was possible in Latin American settings. Replication of our methods and comparison to our findings would improve the validity of the results from the current study. We also recognize that offender population size estimates will be required for many more cities before an inference can be established between offender population size and crime rates, as well as the offender population size and the conditions that may explain the differences in it. To date, very few studies have published similar estimates for other cities across the world. In the current study, we have demonstrated how these estimates can be generated and we encourage replication of our study to determine whether our results are generalizable to other settings in Latin America and elsewhere. We also encourage researchers quantifying offending dynamics to give greater consideration to how they examine explanations for variations in crime levels between different settings.

Generating a value for the size of the offender population that actually exists is unlikely. Therefore, some caution is required in the use of the estimates we have generated because they cannot be compared to a ground truth value and checked for their accuracy. Although this is a limitation, this same limitation is observed in ecology where no known actual population of a large animal species can be calculated (Schwarz and Seber 1999) and estimates are relied upon instead. The methods we have used in the current study are the same methods used in studies for estimating animal populations. As the latter area of science is more advanced than that for estimating offender populations, we encourage criminology researchers to review advances in the estimation of animal population sizes to improve the estimates generated for offender population sizes.

The techniques applied in the current research used arrest data to conclude that, if a person had been arrested, it qualified that individual as an offender during the observation period of the study. This, however, can result in errors in the data caused by wrongful arrests and could have been identified by examining the conviction records associated with the arrests. Conviction records were considered as an alternative data source but, upon examination, we identified issues associated with using these data. This included differences that were likely between the various settings of the conviction procedures, as well as the quality of the criminal investigations, that would have an impact on the comparability of the offender-conviction frequency count data for different settings.

In addition to the size of the offender population, the frequency by which a single offender commits crime, and the duration of the offender's criminal career, also has an impact on the level of crime an area experiences (Visher and Roth 1986). It was beyond the scope of the current study to examine these other two dimensions to offending dynamics. In areas where offenders are more prolific in committing crime and commit crime for a longer duration of their lives, a higher level of crime is likely to result. In the current study, we avoided drawing direct inferences between the offender population size and crime levels because of the small sample of cities and, instead, indicate our results are suggestive of such a relationship. Future research that makes more direct inferential comparisons between offender population sizes and crime levels will also need to consider how the frequency of crime commission by offenders, and their criminal careers, influences the crime rates observed.

## 7. Conclusions

High levels of crime have persisted in Latin American settings for many years. Research that has examined the reasons for the high levels of crime in these settings has mainly focused on the influence of macrostructural conditions, such as social inequality

and poverty, social and economic development, governance, and factors associated to institutional capacity. While useful, these studies have not quantified how these macrolevel explanations translate to the dynamics of offending behaviour. Using three techniques for estimating the hidden population using the known population, data on the arrests of offenders were used to estimate the size of the offender population for robberies from the person for a Brazilian city and a Mexican city and compared to an English city. The estimated offender population sizes in the Brazilian and Mexican cities were several times greater (twenty-five times and three times, respectively) than the size of the offender population in the English city. Estimates for the risk of arrest were also calculated, with these being similar in the Brazilian and Mexican cities and substantially lower than that estimated for the English city.

Estimating the size of the offender population is a valuable measure to calculate because it can offer useful insights that help to explain why an area has crime levels that are higher than other areas. It can also be used for monitoring the impact of interventions and changes in policy. Although the sample size that was used in the current study was modest, the results indicate that offender population sizes are likely to be larger in Latin American settings, with this quantification of criminal behavior adding to our understanding about why these settings experience higher levels of crime.

**Author Contributions:** Conceptualization, S.P.C.; Data curation, S.P.C. and D.L.L.; Formal analysis, S.P.C. and D.L.L.; Investigation, S.P.C. and D.L.L.; Methodology, S.P.C. and D.L.L.; Project administration, S.P.C. and D.L.L.; Resources, S.P.C.; Supervision, S.P.C.; Validation, S.P.C. and D.L.L.; Visualization, S.P.C.; Writing—original draft, S.P.C. and D.L.L.; Writing—review & editing, S.P.C. and D.L.L. Both authors have read and agreed to the published version of the manuscript.

**Funding:** This research received no external funding.

**Institutional Review Board Statement:** Not applicable.

**Informed Consent Statement:** Not applicable.

**Data Availability Statement:** Due to the sensitive personal nature of the content of the offender data used in the current study, these data are not available for public dissemination.

**Conflicts of Interest:** The authors declare no conflict of interest.

## Notes

[1]    Latin America consists of 20 countries and 14 dependent states, such as Puerto Rico.

[2]    We did not use this technique in the current study because it was published once our study had been completed.

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
