# Peer review of "More Offenders, More Crime: Estimating the Size of the Offender Population in a Latin American Setting"

_socsci, doi:10.3390/socsci10090348_

Round 1

Reviewer 1 Report

Page 2, line 65 “data was” to “data were”

Page 2, line 83 – wording -  “In the next section elaborate on the objectives of the current study and describe approaches for estimating…”

Page 3, line 122 “offenders that are caught” to “offenders who were arrested”

Page 3, line 123 “offenders who were not arrested”

                    Line 122 and 123: “that” to “who” and need to clarify “caught” – detained, arrested, prosecuted, sentenced, incarcerated,  etc. – Line 126 uses “arrested”

Page 3, line 123 Arrest data “provide”, not “provides”

Page 3, line 142 & 143 “that” to “who”

Page 3, line 168 “data is” to “data are”

Page 5, line 209 “data that is” to “data that are”

Page 7, line 278 What formula brought us from tables 1 and 2 to table 3,. Fantastic  explanation starting on page 12 does not help the reader on page 7. Maybe short comment about detailed explanation later or move formulas and explanation here.   

Page 7, line 280 “that were active…” to “who were active…”

Page 7, line 285 “population of 684…” Table 2 notes 694

Page 7 area – needs clarification about why there are for Cuiaba (for example) 2057 reported offenses and an estimated 7,567 offenders – There are three times more predicted offenders than reported offenses. Crimes like robbery do not appear to go under reported by 70%, although there are dark figures. Addressed somewhat on  page 10, line 385. Recommend clarifying in the area around page 7 instead of three pages later.

Page 7, line 300 offenders who are

Page 7, line 301 offenders who were

Page 9, line 362 data were available

Page 10, line 397 offenders who were

Author Response

Reviewer 1

Reply 1.1. We thank you for reviewing the article. Thanks for your positive comments and advice on how we can further improve the paper. We have made the changes you have suggested and provide details below.

R1.2. Page 2, line 65 “data was” to “data were”

Reply 1.2. Corrected

R1.3. Page 2, line 83 – wording -  “In the next section elaborate on the objectives of the current study and describe approaches for estimating…”

Reply 1.3. Corrected

R1.4. Page 3, line 122 “offenders that are caught” to “offenders who were arrested”

Reply 1.4. Corrected

R1.5. Page 3, line 123 “offenders who were not arrested”

Reply 1.5. Corrected

R1.6. Line 122 and 123: “that” to “who” and need to clarify “caught” – detained, arrested, prosecuted, sentenced, incarcerated,  etc. – Line 126 uses “arrested”

Reply 1.6. Corrected. Thanks for these suggestions. In places we had previously referred to offenders as being ‘caught’ to be consistent with the terms used when initially explaining capture-recapture models. However, on reflection we agree that changing the text to ‘arrested’ when referring to offenders is clearer.

R1.7. Page 3, line 123 Arrest data “provide”, not “provides”

Reply 1.7. Corrected

R1.8. Page 3, line 142 & 143 “that” to “who”

Reply 1.8. Corrected

R1.9. Page 3, line 168 “data is” to “data are”

Reply 1.9. Corrected

R1.10. Page 5, line 209 “data that is” to “data that are”

Reply 1.10. Corrected

R1.11. Page 7, line 278 What formula brought us from tables 1 and 2 to table 3,. Fantastic  explanation starting on page 12 does not help the reader on page 7. Maybe short comment about detailed explanation later or move formulas and explanation here.

Reply 1.11. Thank you for your comments about the explanation we provide on the methods that starts from page 12. Yes, rather oddly the journal format requires us to describe ‘Materials, Methods and Limitations’ after the Results! We had provided a statement on page 2, from line 85 stating this: “To conform with journal article structure, the data and methods we use are presented in a later section (Materials, Methods and Limitations), and after the presentation and discussion of the results.” However, our preference is to describe data and methods before the results, and as is normal with most articles.

We have reviewed other articles recently published in Social Sciences and observe that several of these articles describe their data and methods before their results. The point you raise is similar to a point raised by another reviewer, therefore, we have changed the order of the paper by moving our description of data and methods to a section before the results, and describing limitations in their own section after the discussion section. By doing so, we believe the revised paper makes the presentation of the results clearer for the reader. We have also written to the journal editor stating we have made these changes in section order in response to the reviews.

R1.12. Page 7, line 280 “that were active…” to “who were active…”

Reply 1.12. Corrected

R1.13. Page 7, line 285 “population of 684…” Table 2 notes 694

Reply 1.13. Corrected. 694 is correct.

R1.14. Page 7 area – needs clarification about why there are for Cuiaba (for example) 2057 reported offenses and an estimated 7,567 offenders – There are three times more predicted offenders than reported offenses. Crimes like robbery do not appear to go under reported by 70%, although there are dark figures. Addressed somewhat on  page 10, line 385. Recommend clarifying in the area around page 7 instead of three pages later.

Reply 1.14. Thank you for this particular observation. We have revised the manuscript in the following ways.

On page 7: “We also note that the estimated size of the offender population in Cuiabá and Escobedo was greater than the number of robbery offenses that were recorded by the police for the time period of analysis (see Table 1). We examine these results further in the discussion section when we consider other research that has examined differences in the dark figure of crime (i.e., the difference between police recorded crime and actual occurrences of crime) between Latin American countries and the United Kingdom.”

In the Discussion section: “We noted in the Results section that the number of robberies recorded by the police in Cuiabá and Escobedo was less than the estimated size of the offender population for this type of crime. Expanding further on research findings about the dark figure of crime, it has been estimated that this figure for robberies in countries in Latin America was between 94% to 96% in 2014 (Jaitman and Anauati 2020) in comparison to a dark figure for robberies of 67% for the United Kingdom in the same study. This suggests that if only 4% to 6% of robberies were reported to the police in Cuiabá and Escobedo, the number of robberies that were actually committed by offenders in each city was more likely to be about 40,000 and 20,000 robberies respectively. Hence, the estimated number of offenders calculated for each city was not greater than the likely number of robbery offenses that were committed in each city.”

R1.15. Page 7, line 300 offenders who are

Reply 1.15. Corrected

R1.16. Page 7, line 301 offenders who were

Reply 1.16. Corrected

R1.17. Page 9, line 362 data were available

Reply 1.17. Corrected

R1.18. Page 10, line 397 offenders who were

Reply 1.18. Corrected

Reviewer 2 Report

Dear author(s),

Your paper was a very interesting reading. In my assessment, this is Great, well written study, with a very good review of the current literature which give readers the necessary context, with brevity. I also commend your initiative in using an innovative approach to answer a very difficult question in regards to the size of offender populations.

It was very interesting to learn about the capture-recapture method, and I believe its application to the estimation of offender population is clever. I also really appreciated the usefulness of the method in estimating the average risk of arrest, which can be used to measure difficult concepts such as impunity. That said, the underlying assumption for the method, that “those arrested are consistent to operate in the same manner as those who have not been arrested” (p. 3) is bold. There are also factors which complicate the use of the method for offender populations, as there are several theoretical differences between robbers which could differentiate between their probability of getting caught. I believe you discuss those limitations well enough. I would suggest you emphasize them bit more in your discussion session, including a debate on the implications of violating the assumptions underlying the methods.

My opinion after reading the article is that the method is useful as one estimate, but not as the only estimate of the size of offender populations. A combination of methods seems most appropriate, as one strategy can be used to validate complement the other.

SPECIFIC COMMENTS:

  1. The sentence: “The definition of robbery from the person for police recording pur-199 poses is also consistent across most parts of the world.” (p. 4, line 199), lacks a supporting citation.
  2. I recommend you describe you estimating techniques before presenting your actual results. I realize this is a minor style point, but I had no clarity what was the credibility of the estimates you were discussing until I finished reading the paper.
  3. Table 3 was a bit difficult to read without lines.

Author Response

Reviewer 2

R2.1. Your paper was a very interesting reading. In my assessment, this is Great, well written study, with a very good review of the current literature which give readers the necessary context, with brevity. I also commend your initiative in using an innovative approach to answer a very difficult question in regards to the size of offender populations.

Reply 2.1. We thank you for reviewing the article. Thanks for your positive comments and advice on how we can further improve the paper. We have made the changes you have suggested and provide details below.

R2.2. It was very interesting to learn about the capture-recapture method, and I believe its application to the estimation of offender population is clever. I also really appreciated the usefulness of the method in estimating the average risk of arrest, which can be used to measure difficult concepts such as impunity. That said, the underlying assumption for the method, that “those arrested are consistent to operate in the same manner as those who have not been arrested” (p. 3) is bold. There are also factors which complicate the use of the method for offender populations, as there are several theoretical differences between robbers which could differentiate between their probability of getting caught. I believe you discuss those limitations well enough. I would suggest you emphasize them bit more in your discussion session, including a debate on the implications of violating the assumptions underlying the methods. My opinion after reading the article is that the method is useful as one estimate, but not as the only estimate of the size of offender populations. A combination of methods seems most appropriate, as one strategy can be used to validate complement the other.

Reply 2.3. Thanks for these comments. We recognise that the statement you quote from the manuscript is bold, but this is written in the context of the literature review and citations we provide before this statement and the citation we provide that relates to this statement. In the Discussion section of the revised manuscript we have included the following about the assumptions of capture-recapture models and how the results from estimating offender population size can be validated. This new text also refers to use of alternative methods to generate estimates of offender population size:

“Capture-recapture models have certain assumptions. We discuss these in more detail in the Limitations section but in general any assumptions that a model possesses can undermine the accuracy of the model if there are questions about the validity of the assumptions. One particular assumption associated with capture-recapture models is that those arrested are considered to operate in the same manner as those who have not been arrested, so they are representative of the full population of active offenders. While some studies have examined this assumption, suggesting that the assumption is valid, these studies have been limited in the settings where they have been applied and where other theoretical factors associated with criminal behavior require consideration. For example, none of these studies are based on samples of offenders in Latin American countries. Therefore, it is possible that in different settings, certain assumptions may be less valid which in turn influences the validity of the results that capture-recapture models generate. While we encourage research that compares how offenders behave in different international settings, to validate the estimates of offender population size that can be generated from capture-recapture models we encourage researchers to consider the use of alternative methods, such as cohort studies and self-reporting offending studies that we described in section 2, and new capture-recapture models that are developed. By doing so, this will allow for results that are generated from different techniques to be compared and help examine if the differences in results relate to the assumptions associated with each model.”

R.2.4. The sentence: “The definition of robbery from the person for police recording purposes is also consistent across most parts of the world.” (p. 4, line 199), lacks a supporting citation.

Reply 2.4. Reference added: (UNODC 2017); UNODC. (2017). Robbery - Statistics and Data. Accessed July 25, 2019: https://dataunodc.un.org/crime/robbery

R.2.5. I recommend you describe you estimating techniques before presenting your actual results. I realize this is a minor style point, but I had no clarity what was the credibility of the estimates you were discussing until I finished reading the paper.

Reply 2.5. Yes, rather oddly the journal format requires us to describe ‘Materials, Methods and Limitations’ after the Results. We provided a statement on page 2, from line 85 stating this: “To conform with journal article structure, the data and methods we use are presented in a later section (Materials, Methods and Limitations), and after the presentation and discussion of the results.” However, our preference is to describe data and methods before the results.

We have reviewed other articles recently published in Social Sciences and observe that several of these articles describe their data and methods before their results. The point you raise is similar to a point raised by another reviewer, therefore, we have changed the order of the paper by moving our description of data and methods to a section before the results, and describing limitations in a section after the discussion section. By doing so, we believe the revised paper makes the paper easier to read. We have also written to the journal editor stating we have made these changes in section order in response to the reviews.

R.2.7. Table 3 was a bit difficult to read without lines.

Reply 2.7. We have added lines to Table 3

Reviewer 3 Report

I suggest the authors to consider the following points.

  • Explain the methodology in detail for non-experts to understand easily
  • Set up hypotheses derived from theoretical review
  • Apply the findings to regions other than Latin American setting
  • Make implications meaningful not only for scholars in the field of criminal justice but for general social scientists 

Author Response

Reviewer 3

I suggest the authors to consider the following points.

Reply 3.1. We thank you for reviewing the article. Thanks for your comments and advice on how we can further improve the paper. We have made the changes you have suggested and provide details below.

R3.2. Explain the methodology in detail for non-experts to understand easily

Reply 3.2. The journal, rather oddly, specifies in its directions to authors for article structure that the ‘Materials, Methods and Limitations’ are described after the Results. We provided a statement on page 2, from line 85 stating this: “To conform with journal article structure, the data and methods we use are presented in a later section (Materials, Methods and Limitations), and after the presentation and discussion of the results.” However, our preference is to describe data and methods before the results, and in particular in the current article because we think this makes the article easier to read.

We have reviewed other articles recently published in Social Sciences and observe that several of these articles describe their data and methods before their results. Therefore, we have changed the order of the paper by moving our description of data and methods to a section before the results, and describing limitations in a section after the discussion section. By doing so, we believe the revised paper makes the description of the methods and the presentation of the results easier to read. We have also made some edits to the methods section to make it easier for non-experts to understand, but we have limited these edits because the other reviewers made a particular comments about how well the original ‘materials and methods’ section was written. For example, “Fantastic explanation starting on page 12 [of data, methods and techniques]”. We have also written to the journal editor stating we have made these changes in section order in response to the reviews.

R3.3. Set up hypotheses derived from theoretical review

Reply 3.3. We did originally state in the introduction section our main hypothesis. In the revised manuscript we have modified the text by including the following just before the Data and Methods section:

“Capture-recapture models generate more accurate estimates when applied for estimating the offender population size for types of crime, rather than all crime more generally (Larson et al. 1994). Hence, in the current study, rather than estimating the offender population size for offenders who committed any type of crime, we estimate the offender population size for a specific type of crime – robbery from the person. We selected this type of crime because robbery from the person takes place in circumstances that are similar in most settings and is similar in its commission in these settings (i.e., it involves an offender using the threat or act of violence to steal an item of property from a person that the person is likely to possess e.g., a cell phone or a wallet/purse containing money or bank cards). Additionally, the police recording definitions for robbery from the person are the same in the countries included in the sample (UNODC 2017), and are types of crime that are reported to the police more than many other crime types that involve face-to-face contact, such as assaults (INEGI, 2019; Home Office 2019). In addition, levels of robberies from the person are higher in Latin American settings than they are in North American and European settings (UNODC, 2017), allowing for the study to speculate on whether these differences are associated with differences in offender population size. From this we hypothesize, and through the use of capture-recapture techniques, that the estimated size of the offender population for robbery in a Latin American setting is greater than that observed in a comparable westernized setting where robbery rates are lower.”

Please also note we have moved the text on data and methods from the ‘Materials, Methods and Limitations’ section to a new ‘Data and Methods’ section that appears before the ‘Results’ in responses to comments from other reviewers. This has resulted in some modifications to the first section of the Results section and the creation of a separate ‘Limitations’ section.

R3.4.Apply the findings to regions other than Latin American setting

Reply 3.4. We assume from this comment that the reviewer only requires us to suggest in the discussion section that the findings, and methods, could also be applied to other settings where offender size population estimates have not been calculated. The study, as the title suggests, is a study to estimate offender population size in Latin American countries. So the focus of the preliminary sections, data and results is on Latin American settings. We have added to the discussion section in the following way:

“We also encourage replication of our study in other settings that experience high levels of crime and where estimates of offender population sizes have not be calculated (e.g., countries in Africa) so that these studies can also improve our understanding about variations in offending dynamics and variations in crime levels.”

R3.5. Make implications meaningful not only for scholars in the field of criminal justice but for general social scientists

Reply 3.5. We have added the following to make the findings and implications of more relevance to scholars in the field of social science:

“Additionally, and in the wider context of social science research, we encourage social scientists who examine the influence that structural conditions (e.g., inequality), development and institutional explanations (e.g., government effectiveness) has on society to consider how these factors relate more directly to the behaviour of individuals. This is often lacking in social science research, and in particular in research that straddles between social science and criminal justice. For example, when researchers reveal significant relationships between variables representing structural conditions and variations in crime levels in statistical models, the validation of these associations would be more robust if each of the significant variables they identify in their models is supported with an explanation about how each of these variables influences offending behavioural dynamics such as the size of the offender population.”